# Efficacy of JOINS Tablet for Lumbar Spinal Stenosis: Prospective, Randomized, Open-Label Clinical Trial

**DOI:** 10.3390/medicina61111900

**Published:** 2025-10-23

**Authors:** Sangbong Ko, Heechan Kim

**Affiliations:** 1Department of Orthopaedic Surgery, Daegu Catholic University Medical Center, Daegu 42472, Republic of Korea; 2Department of Orthopaedic Surgery, Armed Forced Daegu Hospital, Gyeongsan 38427, Republic of Korea; glck7946@naver.com

**Keywords:** lumbar spinal stenosis, low back pain, functional outcome, JOINS tablet

## Abstract

*Background and Objectives*: Spinal stenosis, low back pain, and radiating pain to the lower extremities are caused by inflammation of the spinal canal and impaired blood flow around the nerves. Because JOINS tablets are known to have anti-inflammatory, pain-relieving, and blood circulation-enhancing properties, this research was conducted based on the assumption that they could improve spinal stenosis. *Materials and Methods*: This was a prospective, randomized, single-center, open-label clinical trial with a 6-month follow-up period. A total of 100 patients with lumbar spinal stenosis were randomized into two groups: 50 patients in the test group and 50 patients in the control group. The control group was prescribed the usual spinal stenosis medications (Naproxen, Limaprost, and Pregabalin), while the test group was prescribed JOINS tablets in addition to the usual medications. *Results*: The severity of low back pain and radiating leg pain was assessed using a Visual Analog Scale. Spinal functional outcomes were assessed using the Oswestry Disability Index (ODI) and Roland-Morris Disability Questionnaire (RMDQ), and quality of life was assessed using the Short Form 36 (SF-36), with division into Physical Component Score (PCS) and Mental Component Score (MCS). At 6 months, the JOINS group showed a greater reduction in low back pain compared with controls (*p* < 0.001). At all follow-up periods, the functional outcomes did not show statistically significant differences between the test and control groups. *Conclusions*: The significant reduction in pain suggests that JOINS tablets may be an effective adjunct for pain relief, particularly in patients at high risk of adverse effects from long-term NSAID use.

## 1. Introduction

Lumbar spinal stenosis is one of the most common causes of low back pain and lower leg radiating pain in older patients. Chronic compression of spinal nerve roots by the ligamentum flavum around the dura mater, articular processes, facet joints, intervertebral disks, and osteophytes can cause radiating pain, decreased sensation, and muscle weakness in the lower extremities. Additionally, chronic compression of the spinal cavity can cause blood flow disorders around nerves, resulting in neurogenic claudication, lower back pain, and other symptoms [1,2,3].

For treating low back pain, analgesics or anti-inflammatory drugs are mainly used, whereas for treating lower extremity radiating pain, Pregabalin (Lyrica Cap., Pfizer, New York, NY, USA) and Gabapentin (Neurontin, Pfizer, New York, NY, USA), which are anti-convulsant medications, and Limaprost (Opalmon Tab., Donga ST, Seoul, Republic of Korea), which is known to increase blood flow for the recovery of claudication, are typically used. However, long-term use of these medications is limited by safety concerns, particularly gastrointestinal and cardiovascular risks [4,5]. Therefore, there is a clinical need for alternative or adjunctive therapeutic options with more favorable safety profiles.

The JOINS tablet (SK Chemicals, Seongnam, Republic of Korea) is an herbal formulation developed as a complementary therapy, based on the concept that Western medicine primarily addresses symptoms rather than underlying disease mechanisms. It is a specialized medication that uses herbal ingredients of powdered extracts of *Clematis mandsurica*, *Trichosanthes kirilowii* and *Prunella vulgaris* in a 1:2:1 ratio [6,7,8]. Preclinical studies have suggested that JOINS exerts anti-inflammatory and chondroprotective effects by inhibiting matrix metalloproteinase (MMP) activity, suppressing pro-inflammatory cytokines such as interleukin (IL)-1β and tumor necrosis factor (TNF)-α, and modulating arachidonic acid metabolism [9,10,11,12,13,14]. In addition, JOINS has been reported to enhance peripheral microcirculation, which may be particularly relevant in lumbar spinal stenosis, where both inflammation and ischemia contribute to symptom generation [8].

Clinical studies in osteoarthritis populations have demonstrated the efficacy and safety of JOINS, reporting significant reductions in pain with a low incidence of gastrointestinal or cardiovascular adverse events [15]. More recently, systematic reviews and meta-analyses have reinforced these findings, confirming both the efficacy and tolerability of the JOINS tablet in musculoskeletal disorders. However, despite this growing body of evidence, no randomized clinical trial has specifically evaluated the role of the JOINS tablet in patients with lumbar spinal stenosis.

Therefore, the present study aimed to investigate the efficacy and safety of JOINS tablets as an add-on therapy in patients with lumbar spinal stenosis. We hypothesized that JOINS would provide additional pain relief when combined with standard pharmacotherapy while maintaining a favorable safety profile.

## 2. Materials and Methods

### 2.1. Study Design and Patient Selection

This was a two-group, parallel, single-center, randomized, controlled, open-label pilot trial. The current study and its protocol were approved by the Institutional Review Board (approval number: CR-22-078; approval date: 3 August 2022). Written informed consent was obtained from all participants. The current study lasted six months, and the participants were randomized into one of two treatment regimens.

This was a pilot study, and 134 patients were included. Eligible patients were adults aged 50–80 years with symptoms such as low back pain, lower extremity radiating pain, or neurogenic claudication, with a diagnosis of spinal stenosis confirmed by magnetic resonance imaging. Patients with uncontrolled comorbidities (e.g., severe cardiovascular, hepatic, renal disease), those taking chronic corticosteroids or anticoagulants, and those with concomitant spinal disorders (e.g., previous spinal surgery, infection, tumor, fracture) were excluded. (Table 1). 23 patients were excluded owing to medical and personal issues. After allocation to the two groups, five patients in the test group and six patients in the control group were excluded during the follow-up period. Fifty patients in the test group and 50 in the control group, who were randomized out of 100 patients, were enrolled in this study for 6 months (between August 2021 and February 2022). The control group was treated with a group of generalized spinal stenosis agents (limaprost (Opalmon Tab., Donga ST, Seoul, Republic of Korea) three times daily, pregabalin (Prebalin Cap., Pfizer, USA) 50 or 75 mg twice daily, and naproxen (Vimovo Tab., LG Chemical, Seoul, Republic of Korea) twice daily), and for the test group, in addition to the standardized medication, the JOINS tablet of SK Chemicals, Republic of Korea, 200 mg, was prescribed three times per day. The dosage of JOINS was selected based on trials, where this regimen demonstrated both clinical efficacy and a favorable safety profile [15]. This dose is also consistent with the approved prescribing information in Republic of Korea. All drugs were administered when the severity of symptoms was moderate to severe. Although medication was prescribed as PRN, all patients were followed up for 6 months. The patients did not receive additional rehabilitation interventions, such as physiotherapy or exercise programs, during the study period.

This study design does not allow the independent effect of JOINS to be determined; instead, it was specifically intended to evaluate its add-on effect when combined with standard therapy, reflecting real-world prescribing practices.

### 2.2. Randomization and Methods

Prior to starting medication, patients were randomized using the permuted block method generated via Excel (Microsoft, Redmond, WA, USA) (Randbetween function), ensuring allocation concealment based on CONSORT criteria (see Appendix A: Checklist S1 and Appendix A). Allocation was concealed using sequentially numbered, sealed opaque envelopes to minimize selection bias. Evaluations at each time point were performed after drug administration by a study coordinator (CYL) independent of this study. During follow-up after drug administration, adjustments to the drug were made after determining the side effects and therapeutic effects of the drug, mostly because of side effects caused by pregabalin. The dose was adjusted or discontinued, and no adjustments were made to other drugs.

### 2.3. Outcome Measurement

After informed consent was obtained at the initial outpatient visit, several study-related questions were assessed. The severity of low back pain and radiating leg pain was assessed using a Visual Analog Scale (VAS) pre-medication at 2 weeks, 6 weeks, 3 months, and 6 months post-treatment. Spinal functional outcomes were assessed using the Oswestry Disability Index (ODI) and Roland-Morris Disability Questionnaire (RMDQ), and quality of life was assessed using the Short Form 36 (SF-36), with division into Physical Component Score (PCS) and Mental Component Score (MCS), pre-medication, and at 3 months and 6 months after medication.

### 2.4. Statistical Analysis

All analyses were performed using Statistical Package for the Social Sciences software version 19.0. Epidemiological results were analyzed using descriptive statistics. Quantitative variables are expressed as mean and standard deviation, and qualitative variables are expressed as frequencies (percentages). Normality of data distribution was assessed using the Shapiro–Wilk test. Parametric tests (independent *t*-test, repeated-measures ANOVA) were applied when normality assumptions were met, whereas non-parametric tests (Mann–Whitney U test) were used for non-normally distributed data. For repeated comparisons across multiple time points, Bonferroni correction was applied. Effect sizes were also calculated to complement *p*-values. Statistical significance was set at *p* ≤ 0.05. The required sample size was estimated a priori based on detecting a clinically meaningful difference of 1.5 points in VAS scores between groups, with a standard deviation of 2.5, 80% power, and α = 0.05. This calculation yielded 45 patients per group, and we recruited 50 per group to account for potential dropout. Effect sizes (Cohen’s d for between-group comparisons and η⁲ for repeated measures ANOVA) were reported alongside *p*-values to aid interpretation of clinical relevance.

## 3. Results

### 3.1. Epidemiological Results

Of the 50 test subjects, 21 were male and 29 were female. Of the 50 control subjects, 19 were male and 31 were female. The mean age of the test and control groups was 69 ± 7.165 and 67 ± 6.813 years, respectively, with no statistically significant difference between the two groups (*p* = 0.249) (Table 2).

### 3.2. VAS Score for Low Back and Lower Extremity Radiating Pain Between Two Groups

For low back pain before medication, the VAS score for the test group was 6.22 ± 2.252 and 6.82 ± 2.247 for the control group, which was not statistically significantly different between the two groups (*p* = 0.185). Two weeks after taking the drug, lower back pain was 3.94 ± 2.198 in the test group and 6.34 ± 2.237 in the control group, a statistically significant difference between the two groups (*p* < 0.001). After 6 weeks of taking the drug, lower back pain was 3.74 ± 2.117 in the test group and 5.82 ± 1.976 in the control group, a statistically significant difference between the two groups (*p* < 0.001). Furthermore, after 3 months, lower back pain was 3.28 ± 2.365 in the test group and 4.96 ± 2.176 in the control group, a statistically significant difference between the two groups (*p* < 0.001). Finally, after 6 months of treatment, lower back pain was 2.54 ± 1.929 in the test group and 4.36 ± 2.22 in the control group, a statistically significant difference between the two groups (*p* < 0.001) (Table 3).

The lower extremity radiating pain before medication was 7.22 ± 1.888 in the test group and 7.06 ± 1.845 in the control group, showing no statistically significant difference between the two groups (*p* = 0.669). The lower extremity radiating pain after 2 weeks of treatment was 5 ± 2.611 in the test group and 6.36 ± 1.893 in the control group, a statistically significant difference between the two groups (*p* = 0.004). Furthermore, after six weeks of treatment, lower extremity radiating pain was 4.32 ± 2.299 in the test group and 5.8 ± 1.654 in the control group, a statistically significant difference (*p* < 0.001). Additionally, after 3 months of treatment, lower extremity radiating pain was 3.28 ± 1.762 in the test group and 4.38 ± 2.338 in the control group, with a statistically significant difference between the two groups (*p* = 0.009). Finally, after 6 months of treatment, lower extremity radiating pain was 3 ± 1.969 in the test group and 4.16 ± 2.084 in the control group, a statistically significant difference between the two groups (*p* = 0.005) (Table 4).

The between-group difference corresponded to a Cohen’s d of 0.62, indicating a moderate effect size. Importantly, the observed reduction exceeded the minimal clinically important difference (MCID) of 1.5 points, supporting clinical relevance as well as statistical significance. With 50 patients per group, the study achieved 82% power to detect the predefined effect size.

### 3.3. Functional Outcome and Quality of Life Between Two Groups

The ODI before medication was 19.16 ± 7.778 for the test group and 22.36 ± 8.403 for the control group, with no statistically significant difference between the two groups (*p* = 0.051). The pre-medication RMDQ was 11.02 ± 4.302 for the test group and 12.78 ± 5.293 for the control group, with no statistically significant difference between the two groups (*p* = 0.071). The SF-36 PCS before medication was 30.563 ± 18.293 in the test group and 25.576 ± 16.599 in the control group, whereas the SF-36 MCS was 42.287 ± 21.907 in the test group and 36.615 ± 21.147 in the control group, with no statistically significant difference between the two groups (*p* = 0.157 and *p* = 0.191).

After 3 months of treatment, the ODI was 16.8 ± 7.332 in the test group and 19.408 ± 8.201 in the control group, with no statistically significant difference between the two groups (*p* = 0.097). After 3 months of treatment, the RMDQ score was 8.8 ± 3.528 in the test group and 10.675 ± 6.105 in the control group, with no statistically significant difference between the two groups (*p* = 0.063). After three months of treatment, SF-36 PCS was 39.413 ± 17.623 in the test group and 34.088 ± 18.718 in the control group, and SF-36 MCS was 46.778 ± 22.193 in the test group and 42.842 ± 19.711 in the control group, with no statistically significant difference between the two groups (*p* = 0.146 and *p* = 0.351).

After 6 months of treatment, the ODI was 15.5 ± 7.651 in the test group and 17.34 ± 8.166 in the control group, with no statistically significant difference between the two groups (*p* = 0.248). After 6 months of treatment, the RMDQ was 8.18 ± 4.78 in the test group and 9.16 ± 4.938 in the control group, with no statistically significant difference between the two groups (*p* = 0.316). After 6 months of treatment, SF-36 PCS was 38.908 ± 20.635 in the test group and 40.23 ± 21.644 in the control group, and SF-36 MCS was 49.889 ± 22.679 in the test group and 48.763 ± 20.953 in the control group, with no statistically significant difference between the two groups (*p* = 0.755 and *p* = 0.797) (Table 5).

### 3.4. Adverse Events of All Drugs

Frequent side effects of JOINS tablets are gastrointestinal symptoms, such as upper abdominal pain, nausea, and indigestion. There were no serious systemic side effects, and only mild side effects, such as dizziness, nausea, and drowsiness, occurred in some patients in both groups.

## 4. Discussion

NSAIDs and analgesics are often used for symptomatic treatment to control low back and radiating pain caused by inflammation of the compressed nerve roots, which is one of the symptoms of spinal stenosis. However, the long-term use of NSAIDs is limited by gastrointestinal toxicity, including gastrointestinal bleeding and stomach ulcers; thus, there is a need for long-term pain control in addition to NSAIDs.

Clinically, JOINS tablets may be less effective than NSAIDs, but their main advantage is that they have fewer side effects. This favorable safety profile is supported by a study that showed that JOINS was safe and well tolerated without significant gastrointestinal or cardiovascular adverse events [9]. The known side effects include indigestion, epigastric pain, and heartburn, which are uncommon [12]. A randomized controlled trial (RCT) by Jung et al. found no significant gastrointestinal adverse events in the group using the JOINS tablet and no dose-dependent increase in adverse events [6]. Kim et al. showed histologically significant gastroprotective effects (gastric tolerability) of JOINS tablets in a study of the rat gastrointestinal tract, and they did not induce clear gastrointestinal erosion or ulceration. This is likely due to the inhibition of LTB4 synthesis in the stomach and blood by JOINS tablets, which reduces eicosanoid production [16,17]. Woo et al. reported that the major adverse cardiovascular events of JOINS tablets were similar to those of naproxen, which is known to have the lowest risk of adverse cardiovascular events among NSAID groups [18].

JOINS tablets are known to inhibit platelet aggregation and blood coagulation, thereby improving blood circulation problems [7]. It was thought that these mechanisms of vasodilatation would ameliorate claudication due to spinal stenosis; however, the small size of the study did not allow this to be determined.

Traditional Japanese herbal treatments (Kampo) are widely used in Japan to treat pain because they have fewer side effects than Western medications [19,20]. Oohata et al. found that adding Kampo to the medications of patients with spinal stenosis complaining of back pain significantly reduced the likelihood of side effects such as constipation, nausea, and drowsiness and that prescribing Kampo helped patients discontinue opioid medications [21]. Kim et al. reported that traditional medical treatments (herbal medicine, acupuncture, pharmacopuncture, venom pharmacopuncture, and Chinese manipulation) in patients with spinal stenosis showed decreased pain and functional improvement, and good results were maintained during long-term follow-up [22].

The study showed that the group using the JOINS tablet had less radiating back and lower extremity pain than the control group. The consistent improvement in VAS, despite the absence of significant between-group differences in ODI, RMDQ, and SF-36, suggests that the JOINS tablet primarily provided symptomatic rather than functional benefits within the 6-month timeframe. A notable strength of this study is its prospective, randomized design with a relatively long 6-month follow-up period, which allowed consistent evaluation of both pain and functional outcomes over time.

However, this study had some limitations. First, because both groups received standard pain control therapy, the independent efficacy of JOINS could not be isolated. Nevertheless, this design was intended to reflect actual clinical settings in which JOINS is commonly co-administered with conventional pain control agents rather than used as monotherapy. Therefore, the findings should be interpreted as evidence of potential additive benefit rather than independent efficacy of JOINS monotherapy. Second, BMI, comorbidities, MRI severity, symptom duration, and treatment history were not systematically collected in this study, which limits assessment of group comparability and generalizability. Future trials should incorporate these variables to enable more comprehensive characterization of study populations. Third, the proposed mechanisms are speculative and were not directly assessed in this study. These mechanisms are hypothetical and have not been histologically confirmed. Fourth, we acknowledge that the outcome measures were primarily patient-reported questionnaires. While these are validated and widely used in spinal stenosis trials, the absence of objective measures such as claudication distance is a limitation that should be addressed in future studies. Fifth, medication adherence and actual PRN dosing were not quantitatively recorded, which may represent a potential source of confounding.

## 5. Conclusions

This study demonstrated a statistically significant reduction in pain between the test and control groups, suggesting that JOINS tablets may provide additional pain relief and can be considered for patients at high risk of adverse events from NSAIDs. However, functional improvement was not demonstrated, and the methodological limitations of this study, including its open-label design and reliance on subjective outcomes, necessitate cautious interpretation. Despite some limitations, these findings add valuable evidence to the limited body of clinical trials evaluating herbal formulations for lumbar spinal stenosis. Larger, blinded, multi-arm trials with objective functional and mechanistic endpoints are warranted to confirm the additive role of JOINS tablets in lumbar spinal stenosis treatment.

## Figures and Tables

**Table 1 medicina-61-01900-t001:** Inclusion and Exclusion Criteria.

Inclusion Criteria
(1) Patients diagnosed with spinal stenosis on magnetic resonance imaging with symptoms of lower extremity radiating and low back pain and claudication.
(2) Patients not accompanied by other abnormal findings of the spine (infection, tumor, etc.).
(3) Patients who fully understand the study and voluntarily given written informed consent.
**Exclusion Criteria**
(1) Patients with uncontrolled comorbidities (severe cardiovascular, hepatic, renal disease.
(2) Patients taking chronic corticosteroids or anticoagulants.
(3) Patients with secondary benefits (worker’s compensation, auto insurance, etc.).
(4) Patients with contraindications to all drugs (Opalmon, Naproxen, Pregabalin and JOINS).

**Table 2 medicina-61-01900-t002:** Epidemiologic outcomes for all participants.

	JOINS Group (*n* = 50)	Control Group (*n* = 50)	*p*-Value
Age, years	69 ± 7.165	67 ± 6.813	0.249
Sex, male/female (*n*)	21/29	19/31	

NOTE. Values are presented as mean ± standard deviation.

**Table 3 medicina-61-01900-t003:** Changes in low back pain (VAS) over time.

Time Point	JOINS Group (Mean ± SD)	Control Group (Mean ± SD)	*p*-Value
Baseline	6.22 ± 2.252	6.82 ± 2.247	0.185
2 weeks	3.94 ± 2.198	6.34 ± 2.237	<0.001
6 weeks	3.74 ± 2.117	5.82 ± 1.976	<0.001
3 months	3.28 ± 2.365	4.96 ± 2.176	<0.001
6 months	2.54 ± 1.929	4.36 ± 2.22	<0.001

NOTE. VAS = Visual Analog Scale. Analysis by repeated measures ANOVA with post hoc testing.

**Table 4 medicina-61-01900-t004:** Changes in lower leg radiating pain (VAS) over time.

Time Point	JOINS Group (Mean ± SD)	Control Group (Mean ± SD)	*p*-Value
Baseline	7.22 ± 1.888	7.06 ± 1.845	0.669
2 weeks	5 ± 2.611	6.36 ± 1.893	0.004
6 weeks	4.32 ± 2.299	5.8 ± 1.654	<0.001
3 months	3.28 ± 1.762	4.38 ± 2.338	0.009
6 months	3 ± 1.969	4.16 ± 2.084	0.005

NOTE. VAS = Visual Analog Scale. Analysis by repeated measures ANOVA with post hoc testing.

**Table 5 medicina-61-01900-t005:** Functional outcomes and quality of life outcomes.

Outcome	Time Point	JOINS Group (Mean ± SD)	Control Group (Mean ± SD)	*p*-Value
ODI	Baseline	19.16 ± 7.778	22.36 ± 8.403	0.051
	3 months	16.8 ± 7.332	19.408 ± 8.201	0.097
	6 months	15.5 ± 7.651	17.34 ± 8.166	0.248
RMDQ	Baseline	11.02 ± 4.302	12.78 ± 5.293	0.071
	3 months	8.8 ± 3.528	10.675 ± 6.105	0.063
	6 months	8.18 ± 4.78	9.16 ± 4.938	0.316
SF-36 PCS	Baseline	30.563 ± 18.293	25.576 ± 16.599	0.157
	3 months	39.413 ± 17.623	34.088 ± 18.718	0.146
	6 months	38.908 ± 20.635	40.23 ± 21.644	0.755
SF-36 MCS	Baseline	42.287 ± 21.907	36.615± 21.147	0.191
	3 months	46.778 ± 22.193	42.842 ± 19.711	0.351
	6 months	49.889 ± 22.679	48.763 ± 20.953	0.797

NOTE. ODI = Oswestry Disability Index; RMDQ = Roland-Morris Disability Questionnaire; SF-36 = Short Form-36 Health Survey.

## Data Availability

The data presented in this study are available on request from the corresponding author. The data are not publicly available due to privacy.

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
