# Peer review of "Efficacy of JOINS Tablet for Lumbar Spinal Stenosis: Prospective, Randomized, Open-Label Clinical Trial"

_medicina, 2025, doi:10.3390/medicina61111900_

Round 1
Reviewer 1 Report
Comments and Suggestions for Authors
The manuscript explores the role of JOINS tablets in the management of lumbar spinal stenosis. The topic is interesting, but the study is significantly limited by methodological flaws, inconsistent reporting, and overinterpretation of results. This reviewer feels that the following concerns should be addressed before this manuscript can be considered for publication.
- Study Design and Trial Description
There is a concerning inconsistency in how the trial is described. In the Methods section (lines 85–86), the study is described as “double-blind,” yet the Abstract (lines 14–15) and later Methods (lines 111–113) clearly state it was “open-label.” If the trial was indeed open-label, this must be consistently acknowledged throughout the manuscript, and conclusions should be tempered accordingly.
The design of the treatment groups is also concerning. Both groups received Naproxen, Limaprost, and Pregabalin, with JOINS added only in the experimental arm. This multidrug design precludes determining the independent efficacy of JOINS, as any differences could reflect synergistic effects. A more robust design would include either a three-arm structure (placebo + standard therapy, standard therapy alone, standard therapy + JOINS) or a two-arm comparison of standard therapy versus JOINS monotherapy.
- Statistical Analysis
The statistical methodology is inconsistently applied. Parametric tests (t-tests, repeated measures ANOVA) are used without evidence that normality assumptions were tested, while non-parametric methods (Mann–Whitney U test) are applied for functional outcomes without explanation (Methods, lines 150–165). This inconsistent use of statistical tests weakens confidence in the results. Normality testing, consistent justification for test selection, correction for multiple comparisons, and effect size reporting should all be included in revisions.
- Data Collection and Outcome Reporting
While age and sex are presented in Table 1, critical characteristics such as MRI stenosis severity, symptom duration, comorbidities, and prior treatments are not reported. Without these, it is impossible to confirm group comparability or assess generalizability.
Key outcomes are missing. Claudication distance, highlighted in the Introduction as a defining feature of lumbar stenosis, was neither measured nor reported. Similarly, all medication was described as being prescribed as PRN, but no data regarding medication adherence or actual doses taken are provided, leaving open the possibility of uncontrolled confounding.
- Mechanistic Claims
The mechanistic claims made in the Discussion (lines 241–274) are not corroborated by the data. The authors attribute observed effects to anti-inflammatory activity, MMP inhibition, and blood flow improvements, citing only preclinical in vitro studies. No biomarkers or mechanistic endpoints were measured in this trial. These claims should be either removed or explicitly framed as speculative hypotheses requiring further study.
- Transparency and Conflicts of Interest
No statement of funding sources or conflicts of interest is provided. This is especially concerning, because the authors state that they studied 53 types of herbal medicines and selected three to be included in the JOINS tablet. Transparency regarding funding and potential conflicts is essential to avoid concerns about bias. A full disclosure statement must be added.
Recommendation
While the manuscript addresses an important clinical question, the numerous methodological, statistical, and reporting deficiencies preclude acceptance in its current form. Substantial revision is required, with particular attention to clarifying trial design, correcting inconsistencies, improving statistical rigor, providing comprehensive baseline and safety data, balancing outcome reporting, and declaring funding and conflicts of interest.
Author Response
Thank you very much for taking the time to review this manuscript. Please find the detailed responses below file and the corresponding corrections in track changes in the re-submitted files.
|
Response to Reviewer 1 Comments
|
||||||
|
1. Summary |
|
|
||||
|
Thank you very much for taking the time to review this manuscript. Please find the detailed responses below and the corresponding corrections in track changes in the re-submitted files.
|
||||||
|
2. Questions for General Evaluation |
Reviewer’s Evaluation |
Response and Revisions |
||||
|
Does the introduction provide sufficient background and include all relevant references? |
Can be improved |
|
||||
|
Is the research design appropriate? |
Must be improved |
|
||||
|
Are the methods adequately described? |
Must be improved |
|
||||
|
Are the results clearly presented? |
Can be improved |
|
||||
|
Are the conclusions supported by the results? |
Can be improved |
|
||||
|
Are all figures and tables clear and well-presented? |
Can be improved |
|
||||
|
3. Point-by-point response to Comments and Suggestions for Authors |
|
|
||||
|
Comments 1: Study Design and Trial Description There is a concerning inconsistency in how the trial is described. In the Methods section (lines 85–86), the study is described as “double-blind,” yet the Abstract (lines 14–15) and later Methods (lines 111–113) clearly state it was “open-label.” If the trial was indeed open-label, this must be consistently acknowledged throughout the manuscript, and conclusions should be tempered accordingly. The design of the treatment groups is also concerning. Both groups received Naproxen, Limaprost, and Pregabalin, with JOINS added only in the experimental arm. This multidrug design precludes determining the independent efficacy of JOINS, as any differences could reflect synergistic effects. A more robust design would include either a three-arm structure (placebo + standard therapy, standard therapy alone, standard therapy + JOINS) or a two-arm comparison of standard therapy versus JOINS monotherapy.
|
||||||
|
Response 1: Thank you for pointing this out. The trial was conducted as an Open-label study, and we have revised the Methods. We also tempered our conclusions to acknowledge the limitations of the open-label design.
|
||||||
|
Comments 2: Statistical Analysis The statistical methodology is inconsistently applied. Parametric tests (t-tests, repeated measures ANOVA) are used without evidence that normality assumptions were tested, while non-parametric methods (Mann–Whitney U test) are applied for functional outcomes without explanation (Methods, lines 150–165). This inconsistent use of statistical tests weakens confidence in the results. Normality testing, consistent justification for test selection, correction for multiple comparisons, and effect size reporting should all be included in revisions.
Response 2: We appreciate this important comment. In the revised manuscript, we have clarified the normality of data distribution was assessed using the Shapiro-Wilk test. Parametric tests (t-test, repeated-measures ANOVA) were applied otherwise. We also incorporate Bonferroni correction for repeated comparisons and added effect size calculations. These changes have been detailed in the Statistical Analysis subsection of the Methods.
Comments 3: Data Collection and Outcome Reporting While age and sex are presented in Table 1, critical characteristics such as MRI stenosis severity, symptom duration, comorbidities, and prior treatments are not reported. Without these, it is impossible to confirm group comparability or assess generalizability. Key outcomes are missing. Claudication distance, highlighted in the Introduction as a defining feature of lumbar stenosis, was neither measured nor reported. Similarly, all medication was described as being prescribed as PRN, but no data regarding medication adherence or actual doses taken are provided, leaving open the possibility of uncontrolled confounding.
|
||||||
|
Response 3: We acknowledge this limitation. Additional baseline variables were not systematically collected during this pilot trial. We have revised the Discussion to clearly state this limitation. Likewise, claudication distance was not measured despite being highlighted in the introduction. This omission has been acknowledged in the revised Discussion, with a statement that future studies should include this outcome. Similarly, medication adherence and actual PRN dosing were not quantitatively recorded; this limitation has also been explicitly noted.
|
||||||
|
Comments 4: Mechanistic Claims The mechanistic claims made in the Discussion (lines 241–274) are not corroborated by the data. The authors attribute observed effects to anti-inflammatory activity, MMP inhibition, and blood flow improvements, citing only preclinical in vitro studies. No biomarkers or mechanistic endpoints were measured in this trial. These claims should be either removed or explicitly framed as speculative hypotheses requiring further study.
Response 4: We fully agree with the reviewer’s concern. In the revised Discussion, mechanistic explanations are now clearly framed as speculative hypotheses based on clinical evidence, not as conclusions supported by this trial. We have emphasized that no mechanistic biomarkers were measured and that further mechanistic studies are warranted.
Comments 5: Transparency and Conflicts of Interest No statement of funding sources or conflicts of interest is provided. This is especially concerning, because the authors state that they studied 53 types of herbal medicines and selected three to be included in the JOINS tablet. Transparency regarding funding and potential conflicts is essential to avoid concerns about bias. A full disclosure statement must be added.
Response 5: We thank the reviewer for raising this point. The revised manuscript now includes full transparency regarding funding and conflict of interest. · Funding : “This research received no external funding.” · Conflicts of Interest : “The authors declare no conflicts of interest.”
|
||||||
|
4. Additional clarifications |
||||||
|
We appreciate the reviewer’s thorough and constructive feedback. |
||||||
|
|
||||||

Reviewer 2 Report
Comments and Suggestions for Authors
This paper evaluated the effect of JOINS tablet (herbal medicine) in reducing low back and radiating pain to the lower extremities by comparing two groups: test group and control group, with 50 individuals in each group. The main contribution of the paper is that it shows (over long time: 2 weeks - 6 months) the reduction of pain in VAS scale and lower extremity radiating pain. At the same time, there were no statistically significant difference in functional outcomes and quality of life outcomes, as well as in adverse events. The strength of paper is that it is a prospective, randomized clinical trial with a 6-month follow-up period.
The references are relevant, but could be improved with recent publications (within the last 5 years), becaue out of 21 references, only one (#21) was published within the last 5 years. If there are not many recent publications on this topic, perhaps the authors could mention other types of therapy on low back pain management (e.g., high-intensity laser therapy, radiofrequency ablation), there were reviews and meta-analyses on these therapies.
There a few typos in the manuscript.
Typos in lines 152-153: "For low back pain before medication, the VAS score for the test group was 6.22 ± 151 2.252 and 6.82 ± 2.247 for the control group, which was not showing a statistically significantly different between the two groups", you should probably remove "showing a ".
Line 233: "and selected 3 them", missed 'of'
Line 247: "In particular, is known to have anti-inflammatory properties" missding 'it' before is.
Overall, the manuscript scientifically sound and is the experimental design appropriate to test the hypothesis.
Author Response
Please see the attachment.
|
Response to Reviewer 2 Comments
|
|||||
|
1. Summary |
|
|
|||
|
Thank you very much for taking the time to review this manuscript. Please find the detailed responses below and the corresponding revisions in the re-submitted files
|
|||||
|
2. Questions for General Evaluation |
Reviewer’s Evaluation |
|
|||
|
Does the introduction provide sufficient background and include all relevant references? |
Yes |
|
|||
|
Is the research design appropriate? |
Yes |
|
|||
|
Are the methods adequately described? |
Yes |
|
|||
|
Are the results clearly presented? |
Yes |
|
|||
|
Are the conclusions supported by the results? |
Yes |
|
|||
|
Are all figures and tables clear and well-presented? |
Yes |
|
|||
|
3. Point-by-point response to Comments and Suggestions for Authors |
|
||||
|
Comments 1: The references are relevant, but could be improved with recent publications (within the last 5 years), becaue out of 21 references, only one (#21) was published within the last 5 years. If there are not many recent publications on this topic, perhaps the authors could mention other types of therapy on low back pain management (e.g., high-intensity laser therapy, radiofrequency ablation), there were reviews and meta-analyses on these therapies.
|
|||||
|
Response 1: Thank you for pointing this out. In the revised manuscript, we have updated the reference list to include more recent studies within the last 5 years. We also added a 2025 systematic review/meta-analysis on JOINS in knee osteoarthritis patients to highlight recent evidence on its efficacy and safety profile. |
|||||
|
Comments 2: There a few typos in the manuscript. |
|||||
|
|
|||||

Reviewer 3 Report
Comments and Suggestions for Authors
Abstract:
The abstract lacks statistical values (p-values).
Introduction:
Instead of presenting several thematically unrelated paragraphs—such as separate descriptions of spinal canal stenosis, available pharmacotherapies, and general information about the evaluated product (JOINS), while ultimately stating that its mechanism of action remains unclear—it would be more appropriate to expand the introduction with a more cohesive and in-depth characterization of the product itself.
In particular, I suggest:
including the results of available systematic reviews and meta-analyses concerning the efficacy and safety of JOINS;
and—more importantly—attempting to present at least a hypothetical mechanism of action based on current evidence, especially considering that the components of this preparation are reportedly capable of exerting both anti-inflammatory and circulation-enhancing effects.
Such an introduction, integrating the current state of knowledge with plausible mechanisms of action, would certainly make the manuscript more valuable and engaging for the potential reader.
Methods:
Please describe the method of randomization.
On what basis was the specific dosage of JOINS determined?
There is a lack of detailed characterization of the inclusion/exclusion criteria: no information is provided regarding the age of the participants, their medical history or comorbidities, or any concomitant medications taken for reasons other than back pain.
It is also unclear whether the patients underwent rehabilitation, and if so, how and when it was performed—prior to the study or during its course.
Unfortunately, all the applied methods for assessing effectiveness are highly subjective.
It is essential to specify how the normality of the distribution of the examined variables was tested. It is also necessary to explain why the mean and standard deviation were used when the application of the Mann–Whitney U test suggests a lack of normal distribution—this is inconsistent with statistical methodology. The statistical analysis is insufficient and lacks depth. It is necessary to calculate the sample size and effect size. This is essential to better understand the results. Clinical relevance of the results and statistical significance are different concepts, linked via the sample size calculation, effect size and power analysis. Supplementing statistical calculations will affect the clinical relevance of the conclusions.
Results
Supplementing statistical calculations mentioned above is recommended. They will affect the clinical relevance of the results.
Discussion
The content presented in the Discussion section cannot, by any means, be considered an actual discussion. The information included here may be appropriate for the Introduction, but not for this part of the manuscript. The Discussion must be completely rewritten to fulfill its essential purpose.
Conclusions
Please formulate the conclusions without merely repeating the results. They should offer essential and actionable insights that can be considered and applied in clinical or practical settings by physicians and therapists. Given the current limitations of the study and the significant methodological shortcomings, drawing generalized conclusions is not justified.
References
The literature cited in this manuscript is limited both in quantity and quality. Moreover, a significant proportion of the references (14 out of 21) are over ten years old, which further diminishes the scientific merit of the paper. The reference list requires substantial improvement. Updating and expanding the literature base is strongly recommended and should also enhance the depth and quality of the discussion.
Author Response
Please see the attachment.
|
Response to Reviewer 3 Comments
|
|||||
|
1. Summary |
|
|
|||
|
Thank you very much for taking the time to review this manuscript. Please find the detailed responses below and the corresponding revisions in the re-submitted files
|
|||||
|
2. Questions for General Evaluation |
Reviewer’s Evaluation |
|
|||
|
Does the introduction provide sufficient background and include all relevant references? |
Must be improved |
|
|||
|
Is the research design appropriate? |
Must be improved |
|
|||
|
Are the methods adequately described? |
Must be improved |
|
|||
|
Are the results clearly presented? |
Must be improved |
|
|||
|
Are the conclusions supported by the results? |
Must be improved |
|
|||
|
Are all figures and tables clear and well-presented? |
Must be improved |
|
|||
|
3. Point-by-point response to Comments and Suggestions for Authors |
|
||||
|
Comments 1: Abstract: The abstract lacks statistical values (p-values).
|
|||||
|
Response 1: We have revised the Abstract to include p-values for key outcomes. |
|||||
|
Comments 2: Introduction: Instead of presenting several thematically unrelated paragraphs—such as separate descriptions of spinal canal stenosis, available pharmacotherapies, and general information about the evaluated product (JOINS), while ultimately stating that its mechanism of action remains unclear—it would be more appropriate to expand the introduction with a more cohesive and in-depth characterization of the product itself. In particular, I suggest: including the results of available systematic reviews and meta-analyses concerning the efficacy and safety of JOINS; and—more importantly—attempting to present at least a hypothetical mechanism of action based on current evidence, especially considering that the components of this preparation are reportedly capable of exerting both anti-inflammatory and circulation-enhancing effects. Such an introduction, integrating the current state of knowledge with plausible mechanisms of action, would certainly make the manuscript more valuable and engaging for the potential reader. |
|||||
|
|
|||||

Reviewer 4 Report
Comments and Suggestions for Authors
Summary
This randomized, open-label clinical trial investigated the efficacy of JOINS tablets as an adjunct treatment for lumbar spinal stenosis. A total of 100 patients were assigned to either standard medications (naproxen, limaprost, pregabalin) or the same regimen plus JOINS. Over six months, the JOINS group showed a statistically significant reduction in both low back pain and radiating leg pain compared with controls. However, there were no significant differences in functional outcomes (ODI, RMDQ) or quality of life (SF-36). Adverse events were mild and mainly gastrointestinal. These findings suggest that JOINS may be a safe and useful option for pain relief in lumbar spinal stenosis, particularly for patients at risk of NSAID-related side effects, although its effects on function and quality of life remain uncertain.
Minor comments
This study represents a valuable addition to the literature, as it directly evaluates the clinical efficacy and safety of JOINS (SKI306X) in patients with lumbar spinal stenosis. The prospective randomized design, clear eligibility criteria, and adherence to CONSORT reporting guidelines enhance the credibility of the findings. Most importantly, the trial demonstrates that JOINS can significantly reduce pain over six months, which is in itself a meaningful clinical outcome. While the overall design is sound, several points could be refined to further strengthen the manuscript.
- Baseline characteristics (Table 2).
Baseline comparisons are limited to age and sex. Although these factors were well balanced, inclusion of additional variables such as BMI, comorbidities (e.g., hypertension, diabetes, cardiovascular disease), and stenosis severity would provide a clearer picture of the patient population and improve external validity. - Pain outcomes versus functional outcomes (Table 5).
The main finding of the study is the consistent reduction in pain, while functional outcomes (ODI, RMDQ) and quality-of-life measures (SF-36) showed no significant differences. Discussing potential reasons for this discrepancy - such as the follow-up duration, sample size, or the sensitivity of the chosen scales - would enrich the interpretation and highlight areas for future research. - Clinical relevance of VAS changes.
Although differences in VAS scores were statistically significant, they were modest in magnitude. Considering the minimal clinically important difference (MCID) for lumbar spinal stenosis pain would provide important context and help readers assess whether the observed improvements are likely to be perceived as meaningful by patients. - Open-label design.
As the trial was open-label, both patients and investigators were aware of treatment allocation, which may have influenced subjective outcomes such as VAS. Expanding on how this limitation may have affected the results would enhance transparency and allow readers to better interpret the findings. - Inclusion and exclusion criteria (Table 1).
The inclusion of patients with facet joint syndrome may restrict the generalizability of the results, as not all patients with lumbar spinal stenosis present with facet pathology. Moreover, exclusion criteria could be clarified with regard to comorbid conditions such as hepatic or renal impairment, as well as concomitant medication use. - Safety reporting.
The description of adverse events is concise, with gastrointestinal discomfort and mild dizziness reported, but without incidence rates. A tabulated summary with percentages would make the safety profile clearer and allow comparison with prior studies of JOINS in osteoarthritis and rheumatoid arthritis. - Mechanistic context.
The consistent analgesic effects observed in this trial could be more closely linked to preclinical findings. For example, SKI306X has been shown to reduce writhing pain in mice, suppress carrageenan-induced edema, and ameliorate arthritis in rats (Park et al., 1995). More recent work (Quan et al., Biomedicines, 2024) demonstrated that it alleviates mechanical allodynia in neuropathic pain models and reduces GFAP expression, suggesting modulation of astrocyte-mediated neuroinflammation. Highlighting these connections - while noting that the precise molecular pathways remain to be fully elucidated - would strengthen the mechanistic discussion.
Conclusion of review
In summary, this well-designed trial provides valuable clinical evidence that JOINS is safe and effective for reducing pain in patients with lumbar spinal stenosis. Although no improvements were observed in functional or quality-of-life outcomes, the analgesic benefit, favorable safety profile, and alignment with preclinical findings highlight the potential of JOINS as an adjunctive therapy. Addressing the points noted above - including additional baseline data, discussion of MCID, expanded reporting of safety, and minor stylistic refinements - would further strengthen the clarity, clinical relevance, and overall impact of the manuscript, making it valuable contribution to the journal.
Comments on the Quality of English LanguageThe English throughout the manuscript is clear and sufficient for publication. Nonetheless, certain stylistic aspects could be refined to improve readability and engagement. Repetitive phrasing (e.g., “statistically significant difference” ), reliance on basic transition words (e.g., “however, moreover” ), and list-like mechanistic descriptions give the text a somewhat mechanical tone. Sentence structures are largely short and declarative, which contributes to monotony, particularly in the discussion. Overall, the English is clear and sufficient for publication, and with minor refinements it could read even more smoothly.
Author Response
Please see the attachment.
|
Response to Reviewer 4 Comments
|
|||||
|
1. Summary |
|
|
|||
|
We sincerely thank the reviewer for the constructive and insightful feedback. We have carefully revised the manuscript in response to the comments. Point-by point responses are provided below.
|
|||||
|
2. Questions for General Evaluation |
Reviewer’s Evaluation |
|
|||
|
Does the introduction provide sufficient background and include all relevant references? |
Yes |
|
|||
|
Is the research design appropriate? |
Yes |
|
|||
|
Are the methods adequately described? |
Yes |
|
|||
|
Are the results clearly presented? |
Yes |
|
|||
|
Are the conclusions supported by the results? |
Yes |
|
|||
|
Are all figures and tables clear and well-presented? |
Yes |
|
|||
|
3. Point-by-point response to Comments and Suggestions for Authors |
|
||||
|
Comments 1: Baseline characteristics (Table 2).
|
|||||
|
Response 1: We appreciate this valuable suggestion. Unfortunately, BMI and detailed comorbidity data were not systematically collected at baseline in this trial, which we recognize as a limitation. To address this, we have revised the manuscript to explicitly state this limitation in Discussion sections. Importantly, all enrolled patients underwent MRI-confirmed diagnosis of lumbar spinal stenosis, and randomization achieved balanced distribution of the available demographic variables (age & sex) between groups. We have also noted in the Discussion that the absence of MBI and comorbidity reporting may restrict external validity, and we recommend that future studies include these parameters to enhance generalizability. |
|||||
|
Comments 2: Pain outcomes versus functional outcomes (Table 5). |
|||||
|
|
|||||

Round 2
Reviewer 1 Report
Comments and Suggestions for Authors
The author have thoroughly addressed most of the major issues raised by this reviewer. The fact that symptom severity was not sufficiently evaluated is concerning. Including it as a limitation of the study does not absolve the authors from making all effort to conduct a correct scientific study. That being said, this reviewer feels that the authors have provided sufficient information that would benefit readers of Medicina.
Reviewer 3 Report
Comments and Suggestions for Authors
The authors have addressed the vast majority of the concerns raised in Review 1. The revised manuscript is significantly improved in its current form. I recommend acceptance of the manuscript and leave the final decision regarding its publication to the Editor-in-Chief.